# Hemoglobin Concentration during Early Pregnancy as an Accurate Predictor of Anemia during Late Pregnancy

**DOI:** 10.3390/nu14040839

**Published:** 2022-02-17

**Authors:** Kiwamu Noshiro, Takeshi Umazume, Rifumi Hattori, Soromon Kataoka, Takashi Yamada, Hidemichi Watari

**Affiliations:** 1Department of Obstetrics and Gynecology, Hokkaido University Graduate School of Medicine, Sapporo 060-8638, Japan; ichigoichie_finalfantasy@hotmail.com (K.N.); watarih@med.hokudai.ac.jp (H.W.); 2Department of Obstetrics and Gynecology, Obihiro-Kosei General Hospital, Obihiro 080-0024, Japan; rrrrrr4756@yahoo.co.jp; 3Department of Obstetrics and Gynecology, Hakodate Central General Hospital, Hakodate 040-8585, Japan; sorokata@hakochu-hp.gr.jp; 4Department of Obstetrics and Gynecology, Japan Community Health Care Organization Hokkaido Hospital, Sapporo 062-8618, Japan; yamatakashi@me.com

**Keywords:** anemia, pregnancy, ferritin, transferrin saturation

## Abstract

It is undetermined which blood variables related to iron storage during the first trimester of pregnancy could efficiently predict anemia occurring during the third trimester. Red blood cell count (RBC), hemoglobin concentration, hematocrit, ferritin, iron, and total iron binding capacity (TIBC) were assessed longitudinally during the first, second, and third trimesters of 231 healthy Japanese women. None of the patients had anemia in the first trimester and none used iron supplementation before the second trimester blood test. Anemia was defined as hemoglobin (Hb) < 11 g/dL for the first trimester and Hb < 10.0 g/dL for the third trimester. Forty-seven (20%) women developed anemia in the third trimester. The first trimester RBC, Hb, hematocrit, and ferritin levels were significantly lower in women with third-trimester anemia than those without anemia. The first trimester hemoglobin level exhibited a greater area under the curve of the receiver operating characteristic curve for prediction of the third trimester anemia than other blood variables; the optimal cut-off (12.6 g/dL) of hemoglobin yielded a sensitivity of 83% (39/47). First trimester hemoglobin levels were significantly better predictors of anemia during the third trimester than the indices of iron storage, including serum iron, ferritin, and TIBC levels.

## 1. Introduction

Anemia during pregnancy is a health problem affecting nearly half of all pregnant women worldwide. High fetal demands for iron renders iron deficiency, which is the most common cause of anemia, with other micronutrient deficiencies contributing less frequently [1]. It is estimated that nearly all pregnant women are iron-deficient to some degree and that more than half of pregnant women in developing countries suffer from anemia during pregnancy [2]. Even in developed countries, the iron stores of most pregnant women are considered deficient [3]. Anemia during pregnancy is associated with an increased risk of preterm delivery [4,5] and may increase the risk of postpartum hemorrhage [6], placental abruption [7], cardiac failure, and related death [8]. Children born to iron-deficient mothers show learning and memory impairments that may persist into adulthood [1,9].

Standard care provided to pregnant Japanese women in the first trimester includes a complete blood count (CBC). A hematological analyzer usually yields red blood cell (RBC) count, hemoglobin concentration (Hb), hematocrit (Ht), RBC mean corpuscular volume (MCV), hemoglobin (MCH), and mean corpuscular hemoglobin concentration (MCHC). Serum markers such as ferritin, iron, total iron binding capacity (TIBC), and transferrin saturation (TSAT) are used to diagnose iron deficiency anemia. It may be beneficial for pregnant women and physicians to know which blood variables in the first trimester can predict anemia occurring in the third trimester. However, to our knowledge, this issue has not been extensively studied. In addition, changes in iron storage status have not been studied in pregnant Japanese women.

Serum ferritin levels, which are important in the diagnosis of iron deficiency, have marked variations in the threshold during pregnancy [10], and careful interpretation is necessary. The UK guideline defines anemia of pregnancy as Hb < 11 g/dL in the first trimester and <10.5 g/dL in the second and third trimesters [11]. The International Federation of Gynecology and Obstetrics (FIGO) recommends an Hb level of >10.0 g/dL at the time of delivery [12]. In this study, we used these definitions for the diagnosis of anemia during pregnancy and investigated which blood variables related to iron storage, including ferritin in the first trimester, predicted anemia in the third trimester.

## 2. Materials and Methods

This study was approved by the Institutional Review Board of the Hokkaido University Hospital (019-0390). All participants gave written consent to the study.

All pregnant women scheduled to give birth between October 2018 and December 2019 at Obihiro-Kosei General Hospital, Hakodate Central General Hospital, and Japan Community Health Care Organization Hokkaido Hospital were invited to participate in this study. All participants donated blood samples at each of the first, second, and third trimesters of pregnancy to determine 10 variables: RBC count, Hb, Ht, MCV, MCH, MCHC, ferritin, iron, TIBC, and TSAT. Women with preterm birth at gestational week < 36, those with twin pregnancies, those diagnosed with medical complications at the time of the current pregnancy (including hypothyroidism, Basedow’s disease, IgA nephropathy, and nephrotic syndrome), those with development of hypertensive disorders of pregnancy in the current pregnancy (including chronic hypertension, gestational hypertension, and preeclampsia), and those with anemia of pregnancy in the first trimester of the current pregnancy were excluded from the present analysis. Anemia of pregnancy was defined as Hb < 11 g/dL in the first-trimester, <10.5 g/dL in the second-trimester [11], and <10.0 g/dL in the third-trimester [12]. Since the third trimester defines a wide range of time from 28 to 40 weeks of gestation, the FIGO recommendation of Hb > 10.0 g/dL at the time of delivery [12] was adopted as the criterion of Hb in the third trimester for this study. An iron preparation containing 100 mg of ferrous iron per tablet was prescribed (one tablet/day) at the discretion of the attending physician based on the results of the blood test in the second trimester. Thus, there were no predefined protocols for prescribing iron preparations after the second trimester in this study.

### 2.1. Biochemical Procedures

RBC count, Hb, Ht, MCV, and MCH in the blood were determined at the laboratories of each hospital. Serum and plasma were stored at –80 °C until assays for the following four blood variables: ferritin, iron, and UIBC were measured using ARCHITECT Ferritin (Abbott Japan, Chiba, Japan), Quick Auto Neo Fe (Shino-Test, Tokyo, Japan), and Quick Auto Neo UIBC (Shino-Test), respectively. TSAT (%) was obtained using the following equation: iron (μg/dL)/TIBC (μg/dL) × 100.

### 2.2. Statistical Methods

Statistical analyses were performed using the JMP Pro16© statistical software package (SAS, Cary, NC, USA). Changes in variables within a group were compared using the post hoc Tukey–Kramer method. Categorical variables were compared using the chi-square test. Receiver operating characteristic curves (ROC) were constructed to determine the optimal cut-off of blood variables for the prediction of anemia of pregnancy occurring in the third trimester. In all analyses, the statistical significance was set at *p* < 0.05.

## 3. Results

A total of 231 women without first trimester anemia were analyzed in this study (Figure 1). None of the participants were prescribed iron supplementation until the blood tests were performed in the second trimester.

### 3.1. Demographic Characteristics

Of the 231 women without first trimester anemia during pregnancy, 36 (16%) and 47 (20%) women developed anemia during the second and third trimesters, respectively (Table 1). One hundred and eighty (78%) patients delivered vaginally. The mean gestational weeks of blood sampling were 10.2, 25.8, and 36.2 for the first, second, and third trimesters, respectively. Fifteen women received iron supplements between the second and third trimesters.

### 3.2. Blood Parameters during Pregnancy

RBC, Hb, Ht, MCH, MCHC, ferritin, iron, and TSAT levels decreased, and TIBC increased from early to late pregnancy. Hb < 10.5 g/dL in the second trimester was found in 36 (16%), and Hb < 10.0 g/dL in the third trimester in 47 (20%). Ferritin levels of <30 ng/mL were found in 214 women (93%) in the second trimester and 221 women (96%) in the third trimester (Table 2).

### 3.3. Anemia Characteristics during First-Trimester

The characteristics of the first trimester in 47 women who developed anemia in the third trimester were investigated (Table 3). The fraction of nulliparas was low in the anemia group at 21% (10/47). RBC, Hb, Ht, MCV, and ferritin were already low, and TIBC was high in the first trimester. In contrast, MCH, MCHC, iron, and TSAT levels did not differ between the two groups. Among the 47 patients with anemia in the third trimester, 20 (43%) had anemia (Hb < 10.5 mg/dL) in the second trimester.

### 3.4. Blood Variables of First Trimester as Predictors of Third Trimester Anemia

The areas under the curve (AUC) of the receiver operating characteristic (ROC) curves were 0.76 for Hb, and 0.75 for Ht, respectively, during first trimester (Table 4), suggesting that first trimester Hb and Ht were better predictors of the third trimester anemia than other blood variables. The optimal cut-offs determined by ROC were 12.6 g/dL for first trimester Hb yielded a sensitivity of 83% (39/47) and a specificity of 59% (108/184), and 36.8% for first trimester Ht yielded a sensitivity of 70% (33/47) and a specificity of 66% (121/184).

## 4. Discussion

The results of this study demonstrated that as many as 20% of healthy Japanese women developed third trimester anemia; most patients with anemia were iron-deficient, and the first trimester Hb was a significantly better predictor of anemia occurring in the third trimester than other indices that were more directly associated with iron storage status.

Iron deficiency is associated with multiple adverse outcomes for both mother and infant, including increased risks of hemorrhage, sepsis, maternal mortality, preterm delivery, low birth weight, and perinatal mortality. The World Health Organization has emphasized the importance of iron deficiency and anemia during pregnancy as a public health problem [1]. In our study, one-fifth of the women with uncomplicated pregnancies developed third trimester anemia, and approximately half of the women without second trimester anemia developed anemia during the third trimester. In this study, the FIGO recommended Hb < 10.0 g/dL at the time of delivery [12] was used as the definition of anemia in the third trimester. Moreover, if the WHO recommended amount of Hb < 11.0 g/dL during pregnancy [1] is considered as anemia, more than half of healthy Japanese women in the third trimester will be anemic. Therefore, early prediction of a higher risk of anemia in pregnant women is clinically important to allow for early intervention, such as iron supplementation.

Serum iron and TIBC levels were used to calculate TSAT. TSAT values < 20% suggest an inadequate supply of iron for hemoglobin synthesis and red cell production [13]. Serum iron represents the iron bound to the transport protein, transferrin, available for incorporation into hemoglobin in developing erythroblasts in the bone marrow [13] and ranges from 4.7–62 μmol/L (26–348 μg/dL) in pregnant women [14]. Serum ferritin is the protein in the plasma that reflects the body’s iron stores under normal conditions [13] and ranges from 3.0–129 ng/mL in pregnant women [14]. In this study, the average second-trimester serum iron level was 71 (9–239) μg/dL, TSAT level was 15% (1.7–68%), and serum ferritin level was 6.7 (1.2–128) ng/mL, respectively (Table 2). Thus, most patients in the second trimester may have been iron deficient.

Iron requirements increase from 0.8 mg/day in the first trimester to 7.5 mg/day in the third-trimester, averaging 4.4 mg/day [15]. Conversely, intestinal iron absorption is tightly controlled and depends on the iron needs of the body [16]. More than 90% of healthy pregnant women have ferritin levels < 30 ng/mL during the second trimester, suggesting that early iron supplementation is important. There is a strong association between moderate to severe anemia at 28 weeks of pregnancy, and the severity of bleeding during and after delivery [17], as well as the prevention of anemia during delivery is important.

The first trimester Hb value was able to efficiently distinguish women at an elevated risk of anemia during the third trimester. Of the 115 women with Hb < 12.6 g/dL during the first trimester, 39 (34%) developed anemia during the third trimester, while only 8 of 116 (6.9%) with Hb ≥ 12.6 g/dL during first trimester developed anemia during the third trimester. This result may provide a rationale for early initiation of iron supplementation even in women without anemia, but with lower first trimester Hb levels, and may contribute to the reduction of anemia frequency.

In clinical practice, iron-related indicators such as ferritin and TSAT may be used as criteria to determine the need for iron supplementation; however, the cost of measuring iron-related indicators has been limiting. In this study, we suggest Hb that can be measured economically could predict anemia in the third trimester of pregnancy and be an important indicator in clinical practice. An early oral iron supplement has been shown to improve iron storage [8,18,19] and reduce the incidence of anemia. Therefore, considering iron supplementation for women with Hb < 12.6 g/dL early in pregnancy may help prevent anemia during labor.

However, it was unclear whether iron supplementation favorably affected pregnancy outcomes due to the limited size of the study population. Despite the high incidence and burden of anemia, there is a paucity of good-quality clinical trials assessing maternal and neonatal effects of iron administration in women with anemia [8].

## 5. Conclusions

This prospective study demonstrated that approximately one-fifth of the participants developed anemia during the third trimester. The first trimester Hb value most efficiently predicted anemia occurring in the third trimester than the indices of iron storage status, such as serum iron, ferritin, and TSAT levels.

## Figures and Tables

**Figure 1 nutrients-14-00839-f001:**
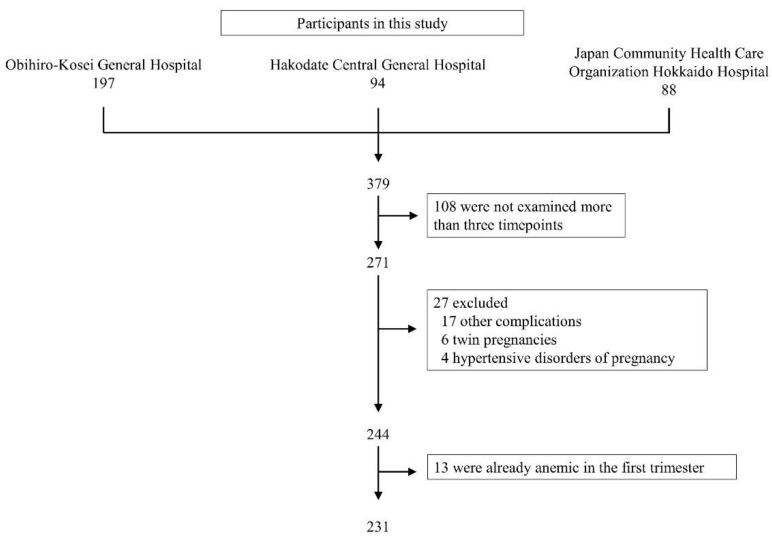
Study design.

**Table 1 nutrients-14-00839-t001:** Demographic characteristics.

Number of women	231
Age, year	32.0 (18–44)
Nulliparous women	82 (35%)
Pre-pregnancy body mass index, kg/m^2^	21.4 (15.8–39.4)
Gestational weeks at delivery	39.1 (36.1–41.7)
Vaginal delivery	180 (78%)
Cesarean section	51 (22%)
Birthweight, g	3086 (2018–4314)
GW at blood sampling	
First trimester	10.2 (7.9–13.9)
Second trimester	25.8 (23.0–29.7)
Third trimester	36.2 (34.6–38.7)
Iron supplementation before third trimester exams	
Number of women	15 (6%)
Total iron dose, mg	2150 (700–4200)
Second trimester anemia (hemoglobin < 10.5 g/dL)	36 (16%)
Third trimester anemia	47 (20%)

Data are presented as the mean (range).

**Table 2 nutrients-14-00839-t002:** Clinical data and blood parameters during pregnancy.

	First Trimester	Second Trimester	Third Trimester
Clinical data			
Maternal body weight, kg	54.0 ± 10.4	58.9 ± 9.9 *	62.9 ± 10.3 *
Systolic blood pressure, mmHg	110 ± 13	108 ± 13	112 ± 13
Diastolic blood pressure, mmHg	64 ± 10	60 ± 9 *	65 ± 10
Weight gain in pregnancy, kg	0.6 ± 2.4	5.5 ± 3.2 *	9.5 ± 4.0 *
Weight gain, %	1.2 ± 4.4	11 ± 6.5 *	19 ± 8.5 *
Iron supplementation before third trimester exams †			
Number of women	0	0	15 (6%)
Total iron dose, mg	0	0	2150 (700–4200)
Blood data			
Red blood cell count, ×10^4^/μL	418 (331–510)	368 (280–457) *	383 (282–472) *
Hemoglobin, g/dL	12.6 (11.0–15.1)	11.2 (8.0–13.3) *	10.8 (8.5–13.4) *
<11.0 g/dL	0	90 (39%)	130 (56%)
<10.5 g/dL	0	36 (16%)	86 (37%)
<10.0 g/dL	0	10 (4%)	47 (20%) ‡
Hematocrit, %	37.2 (31.4–44.7)	33.4 (26.3–39.4) *	32.7 (25.5–40.6) *
MCV, fL	89.6 (71.0–105)	91.7 (76.2–108) *	86.8 (63.9–107) *
MCH, pg	30.3 (22.7–36.0)	30.8 (23.7–37.2)	28.8 (20.6–36.9) *
MCHC, %	33.8 (31.4–36.1)	33.7 (30.3–36.1)	33.0 (28.4–47.1) *
Ferritin, ng/mL	39.2 (1.1–214)	6.7 (1.2–128) *	5.7 (1.1–65.8) *
<30 ng/mL	78 (34%)	214 (93%)	221 (96%)
Iron, μg/dL	110 (16–203)	71 (9–239) *	44 (12–245) *
TIBC, μg/dL	338 (231–517)	469 (274–667) *	549 (348–740) *
Transferrin saturation, %	32 (3.6–85)	15 (1.7–68) *	7.9 (2.2–69.6) *

Clinical data are presented as means (range). Blood data are presented as medians (range). * *p* < 0.05 versus baseline value of the same group determined in the first trimester. †: Iron supplementation was initiated after the blood tests performed in the second trimester and before the third trimester exams. ‡: 47 women with Hb < 10.0 g/dL in the third trimester included 20 women with Hb < 10.5 in the second trimester. MCV: mean corpuscular volume; MCH: mean corpuscular hemoglobin; MCHC: mean corpuscular hemoglobin concentration; TIBC: total iron-binding capacity.

**Table 3 nutrients-14-00839-t003:** Demographic characteristics and blood variables in anemic women.

Clinical Data	Anemia	No Anemia	*p*-Value
No. of women	47	184	
Age (year)	32 (18–44)	32 (19–44)	0.891
Nulliparous women	10 (21%)	72 (39%)	0.026
Pre-pregnancy BMI (kg/m^2^)	20.8 (16.4–30.5)	21.5 (15.8–39.4)	0.255
GW at delivery	39.0 (36.1–41.3)	39.2 (36.1–41.7)	0.385
Vaginal delivery	38 (81%)	142 (77%)	0.695
Cesarean section	9 (19%)	42 (23%)	
Birthweight (g)	3163 (2348–4312)	3067 (2018–4314)	0.131
Hb < 10.5 at second trimester	20 (43%)	16 (9%)	<0.001
Iron supplementation before third trimester exams *			
No. of women	8 (17%)	7 (4%)	0.003
Blood data			
RBC count, ×10^4^/μL	407 (340–510)	421 (331–498)	0.011
Hemoglobin, g/dL	11.9 (11.0–14.2)	12.7 (11.0–15.1)	<0.001
Hematocrit, %	35.4 (31.4–42.7)	37.6 (31.8–44.7)	<0.001
MCV, fL	88.8 (71.0–95.7)	90.0 (72.4–105)	0.021
MCH, pg	30.0 (22.7–33.0)	30.4 (22.7–36.0)	0.078
MCHC, %	34.0 (31.9–36.0)	33.8 (31.4–36.1)	0.418
Ferritin, ng/mL	29.0 (1.9–188)	44.8 (1.1–214)	<0.001
Iron, μg/dL	104 (16–202)	114 (26–203)	0.248
TIBC, μg/dL	361 (237–503)	333 (231–517)	0.028
Transferrin saturation, %	29.5 (3.6–58.6)	32.8 (9.0–84.6)	0.155

Clinical data are presented as the mean (range). Blood data are presented as the median (range). * Iron supplementation was initiated after the blood tests performed in the second trimester and before the third trimester exam. MCV: mean corpuscular volume; MCH: mean corpuscular hemoglobin; MCHC: mean corpuscular hemoglobin concentration; TIBC: total iron-binding capacity.

**Table 4 nutrients-14-00839-t004:** Screening characteristics of optimal cut-offs suggested by Receiver Operating Characteristic for prediction of third trimester anemia.

Blood Variables	AUC	Cut-Off	Sensitivity	Specificity
RBC count, ×10^4^/μL	0.62 *	389	32% (15/47)	86% (159/184)
Hemoglobin, g/dL	0.76	12.6	83% (39/47)	59% (108/184)
Hematocrit, %	0.75	36.8	70% (33/47)	66% (121/184)
MCV, fL	0.61 *	90.0	68% (32/47)	51% (93/184)
MCH, pg	0.58 *	28.7	30% (14/47)	89% (163/184)
MCHC, %	0.54 *	33.7	36% (17/47)	54% (99/184)
Ferritin, ng/mL	0.66 *	33.8	64% (30/47)	64% (117/184)
Iron, μg/dL	0.55 *	83.0	38% (18/47)	77% (141/184)
TIBC, μg/dL	0.60 *	371	51% (24/47)	27% (49/184)
Transferrin saturation (%)	0.57 *	11.6	17% (8/47)	96% (177/184)

* *p* < 0.05 vs. the value of hemoglobin. AUC: area under the curve; MCV: mean corpuscular volume; MCH: mean corpuscular hemoglobin; MCHC: mean corpuscular hemoglobin concentration; TIBC: total iron-binding capacity.

## Data Availability

The data presented in this study are available on request from the corresponding author. The data are not publicly available due to privacy.

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
