# Peer review of "Hemoglobin Concentration during Early Pregnancy as an Accurate Predictor of Anemia during Late Pregnancy"

_nutrients, 2022, doi:10.3390/nu14040839_

Round 1

Reviewer 1 Report

The paper is well written and describes in an appropriate manner a well-designed study of high interest due to its importance to clinical practice: it approaches anemia in pregnant women - an issue highly prevalent and having demonstrated deleterious consequences on both mother and child. The conclusions are fair and have a high benefit/cost ratio in clinical practice.

A minor suggestion: in paragraph 3.3. Anemic characteristics during first-trimester replace the word ANEMIC with ANEMIA

Author Response

Reviewer 1

The paper is well written and describes in an appropriate manner a well-designed study of high interest due to its importance to clinical practice: it approaches anemia in pregnant women - an issue highly prevalent and having demonstrated deleterious consequences on both mother and child. The conclusions are fair and have a high benefit/cost ratio in clinical practice.

A minor suggestion: in paragraph 3.3. Anemic characteristics during first-trimester replace the word ANEMIC with ANEMIA

Response: Thank you for your comments. I revised that phrase.

Reviewer 2 Report

Thus study provides evidence for first-trimester Hb values (not commonly regarded as  indicators of anemia) best  capable to  distinguish women with an elevated risk of developing anemia during third trimester, providing a rationale for initiating iron supplementation earlier than second-third semester. I have only a few minor comments:

1. Abstractconsecutive healthy? UNCLEAR MEANING.

First trimester hemoglobin levels were considered better predictors ....RELATIVELY BETTER (how much better, significantly?)  

2 INTRODUCTION. Serum markers such as ferritin, iron, total iron binding capacity (TIBC), and transferrin saturation (TSAT) may also be used to diagnose iron deficiency anemia.  IRON DEFICIENCY ANEMIA CAN NOT BE DEFINED WITHOUT TSAT-AND TIBC-FERRITIN (in the absence of inflammation). SUGGEST TO STATE ARE USED INSTEAD OF MAYBE USED

 3. METHODS. BIOCHEMICAL PROCEDURES   ferritin, iron and TIBC ....   use here UIBC (from which TIBC is calculated)

4. DISCUSSION. P 7 L 297   detrimental means harmful. write instead LIMITING

Author Response

Reviewer 2

Thus study provides evidence for first-trimester Hb values (not commonly regarded as indicators of anemia) best capable to  distinguish women with an elevated risk of developing anemia during third trimester, providing a rationale for initiating iron supplementation earlier than second-third semester. I have only a few minor comments:

  1. Abstract-  consecutive healthyUNCLEAR MEANING.

First trimester hemoglobin levels were considered better predictors ....RELATIVELY BETTER (how much better, significantly?)  

Response: Thank you for your comments. I tested between the AUC of hemoglobin and of the other variables in Table 4 and added “significantly” in the text.

2 INTRODUCTION. Serum markers such as ferritin, iron, total iron binding capacity (TIBC), and transferrin saturation (TSAT) may also be used to diagnose iron deficiency anemia.  IRON DEFICIENCY ANEMIA CAN NOT BE DEFINED WITHOUT TSAT-AND TIBC-FERRITIN (in the absence of inflammation). SUGGEST TO STATE ARE USED INSTEAD OF MAYBE USED

Response: Thank you for your comments. I revised that phrase.

  1. METHODS. BIOCHEMICAL PROCEDURES   ferritin, iron and TIBC....   use here UIBC (from which TIBC is calculated)

Response: Thank you for your comments. I revised that phrase.

  1. DISCUSSION. P 7 L 297   detrimentalmeans harmful. write instead LIMITING

Response: Thank you for your comments. I revised that phrase.

Reviewer 3 Report

In this study, the authors enrolled pregnant participants and monitor blood parameters in the first, second and third trimester of pregnancy. According to the diagnosis criteria of anemia, they identified 47 anemia participants in the third trimester. By analyzing the blood examination data, they concluded that hemoglobin concentration in the first trimester was able to predict anemia in the third trimester. Bellow, I list my comments and suggestions.

  1. This study is to identify the reliable predictor of anemia of pregnancy. Therefore, the definition of anemia must be clear. In the abstract, anemia was defined as “Hb < 11 g/dL for the first-25 trimester and Hb < 10.0 g/dL for the third-trimester”. However, at line 69, the UK guideline of anemia definition was “Hb < 11 g/dL in the first trimester and < 10.5 g/dL in the second and third trimesters”. It seemed that the definition of anemia diagnosis was not consistent within the manuscript.

  1. The participants were enrolled in three different hospitals. I wonder whether the demographic data and blood examination results of the participants from the three different hospitals varied?

  1. In Table 2, p-value was shown. However, what comparison was p-value generated from? First trimester vs. second one, second trimester vs. third one or first trimester vs. third on? Please clarify this issue and provide thin information in table legend.

  1. According to Table 2, the Hb concentrations of 36 participants were lower than 10.5 g/dL in the second trimester. And 47 participants had Hb lower than 10.5 g/dL in the third trimester. Is the 36 a subset of the 47?

Author Response

Reviewer 3

In this study, the authors enrolled pregnant participants and monitor blood parameters in the first, second and third trimester of pregnancy. According to the diagnosis criteria of anemia, they identified 47 anemia participants in the third trimester. By analyzing the blood examination data, they concluded that hemoglobin concentration in the first trimester was able to predict anemia in the third trimester. Bellow, I list my comments and suggestions.

  1. This study is to identify the reliable predictor of anemia of pregnancy. Therefore, the definition of anemia must be clear. In the abstract, anemia was defined as “Hb < 11 g/dL for the first-25 trimester and Hb < 10.0 g/dL for the third-trimester”. However, at line 69, the UK guideline of anemia definition was “Hb < 11 g/dL in the first trimester and < 10.5 g/dL in the second and third trimesters”. It seemed that the definition of anemia diagnosis was not consistent within the manuscript.

Response: Thank you for your comments. I added the following sentence to the " Materials and Methods ".

Because the third trimester defines a wide range of time from 28 to 40 weeks of gestation, the FIGO recommendation of Hb > 10.0 g/dL at the time of delivery [12], was adopted as the criterion of Hb in the third trimester in this study.

  1. The participants were enrolled in three different hospitals. I wonder whether the demographic data and blood examination results of the participants from the three different hospitals varied?

Response: Thank you for your comments.

Representative data for three hospitals are shown in the table below. Analysis of variance (ANOVA or chi-square test) showed no significant between-group differences. Because there were no differences in the data between hospitals, this study was intended to present only the results of the analysis of the combined data and to clarify the results.

Supplementary table                                                                                  

Hospital                                     A (n=64)                      B (n=69)              C (n=98)                                                   p-value       

age                                             31.5±5.8                       33.0±4.6              31.6±4.6                                                   0.103

pre-pregnancy BMI                   22.0±4.3                       21.0±3.3              21.2±3.3                                                   0.310

Nulliparous women                   45%(29/64)                  26%(18/69)         36%(35/98)                                                                                        0.067

Hemoglobin                              

 1st trimester                       12.8±0.8                       12.4±0.8              12.6±0.8                                                                                        0.060

3rd trimester                       10.9±1.0                       10.7±0.9              11.0±1.1                                                                                        0.181

Ferritin

1st trimester                       51.3±31.8                     41.1±36.6            54.0±40.7                                                                                        0.086

3rd trimester                       7.7±4.7                         10.3±12.2            6.4±6.6                                                                                        0.055     

  1. In Table 2, p-value was shown. However, what comparison was p-value generated from? First trimester vs. second one, second trimester vs. third one or first trimester vs. third on? Please clarify this issue and provide thin information in table legend.

Response: I'm sorry that I didn't clarify the method of the statistical test. Table 2 presented only the results of an analysis of variance (ANOVA). I have added the test of Wilcoxon’s rank-sum test and Student’s t-test with Bonferroni correction to compare variables within a group, and following phrase in table legend “*P<0.05 versus baseline value of the same group determined in the first trimester”.

  1. According to Table 2, the Hb concentrations of 36 participants were lower than 10.5 g/dL in the second trimester. And 47 participants had Hb lower than 10.5 g/dL in the third trimester. Is the 36 a subset of the 47?

Response: Thank you for your comments. 47 women with Hb < 10.0 g/dL in the third trimester included 20 women with Hb < 10.5 in the second trimester. I added that fact to the legend of Table 2.
